# An *EHPB1L1* Nonsense Mutation Associated with Congenital Dyserythropoietic Anemia and Polymyopathy in Labrador Retriever Littermates

**DOI:** 10.3390/genes13081427

**Published:** 2022-08-11

**Authors:** G. Diane Shelton, Katie M. Minor, Ling T. Guo, Alison Thomas-Hollands, Koranda A. Walsh, Steven G. Friedenberg, Jonah N. Cullen, James R. Mickelson

**Affiliations:** 1Department of Pathology, School of Medicine, University of California San Diego, La Jolla, CA 92093, USA; 2Department of Veterinary and Biomedical Sciences, College of Veterinary Medicine, University of Minnesota, Saint Paul, MN 55108, USA; 3Department of Clinical Sciences and Advanced Medicine, School of Veterinary Medicine, University of Pennsylvania, Philadelphia, PA 19104, USA; 4Department of Clinical Pathobiology, University of Pennsylvania School of Veterinary Medicine, Philadelphia, PA 19104, USA; 5Department of Veterinary Clinical Sciences, College of Veterinary Medicine, University of Minnesota, Saint Paul, MN 55108, USA

**Keywords:** dog, muscle, anemia, myopathy, whole genome sequencing, animal model

## Abstract

In this report, we describe a novel genetic basis for congenital dyserythropoietic anemia and polymyopathy in Labrador Retriever littermates characterized by incidental detection of marked microcytosis, inappropriate metarubricytosis, pelvic limb weakness and muscle atrophy. A similar syndrome has been described in English Springer Spaniel littermates with an early onset of anemia, megaesophagus, generalized muscle atrophy and cardiomyopathy. Muscle histopathology in both breeds showed distinctive pathological changes consistent with congenital polymyopathy. Using whole genome sequencing and mapping to the CanFam4 (*Canis lupus familiaris* reference assembly 4), a nonsense variant in the *EHBP1L1* gene was identified in a homozygous form in the Labrador Retriever littermates. The mutation produces a premature stop codon that deletes approximately 90% of the protein. This variant was not present in the English Springer Spaniels. Currently, *EHPB1L1* is described as critical to actin cytoskeletal organization and apical-directed transport in polarized epithelial cells, and through connections with Rab8 and a BIN1-dynamin complex generates membrane vesicles in the endocytic recycling compartment. Furthermore, *EHBP1L1* knockout mice die early and develop severe anemia. The connection of EHBP1L1 to BIN1 and DMN2 functions is particularly interesting due to *BIN1* and *DMN2* mutations being causative in forms of centronuclear myopathy. This report, along with an independent study conducted by another group, are the first reports of an association of *EHBP1L1* mutations with congenital dyserythropoietic anemia and polymyopathy.

## 1. Introduction

Congenital dyserythropoietic anemia and congenital polymyopathies are rare, inherited disorders. A clinical syndrome of congenital dyserythropoiesis and polymyopathy was first reported in 1991 in young, related male and female English Springer Spaniel (ESS) dogs [1], and thirty years later in young male Labrador Retriever littermates [2]. The clinical presentation differed between the two breeds. An early onset of anemia, megaesophagus, generalized muscle atrophy and cardiomyopathy was observed in the English Springer Spaniels while an incidental detection of marked microcytosis, inappropriate metarubricytosis, pelvic limb weakness and muscle atrophy without cardiomyopathy were observed in the Labrador Retrievers. Despite the variable clinical presentations, similar changes were identified in erythrocyte morphology and muscle histopathology in both breeds [1,2].

Several congenital dyserythropoietic anemias occur in people [3], while in dogs congenital anemias are more commonly associated with enzyme deficiencies [4]. In both people [5] and dogs [6], congenital myopathies are defined as a heterogeneous group of non-dystrophic neuromuscular disorders classified based on predominant, usually structural, pathological findings in muscle biopsy samples. Both X-linked myotubular myopathy (XLMTM) [7,8] and centronuclear myopathy (CNM) [9] have been described in Labrador Retrievers and mutations identified [8,9]. Of note, pathological and histochemical changes in muscle biopsies are very similar between XLMTM, CNM and the changes reported in the Labrador Retrievers and ESS with dyserythropoietic anemia and polymyopathy [1,2]. Such changes include excessive variability in myofiber size, central nuclei and central accumulations of oxidative activity. Anemia and abnormal erythrocyte morphology has not been reported in any of the published cases of XLMTM or CNM [7,8,9].

Investigations into the molecular basis for this clinical syndrome of dyserythropoietic anemia and polymyopathy in dogs were not performed at the time of the original publications. Here, we use whole genome sequencing (WGS) and mapping of a dog reference genome assembly to identify a likely genetic basis for this syndrome in the young Labrador Retrievers.

## 2. Materials and Methods

### 2.1. Animals

The two affected Labrador Retriever littermates previously reported were evaluated further in this study. The clinical history and pathological changes in erythrocytes and muscles from both dogs were previously described [2]. A brief review of the pathological changes in erythrocytes and in myofibers is presented (Figure 1) to facilitate an understanding for the non-pathologist. Changes typical of dyserythropoiesis in red blood cells and bone marrow are demonstrated (Figure 1A–D) and the altered morphology typical of a congenital myopathy is illustrated (Figure 1E,F).

Genomic DNA from one of the Labrador Retriever littermates was isolated from an archived frozen diagnostic muscle biopsy; the other was isolated from buccal swabs, both using the Gentra PureGene blood kit (Qiagen, Germantown, MD, USA). DNA from affected related ESS dogs originally reported in 1991 [1] was isolated from archived paraffin blocks using the PureGene kit with manufacturer’s instructions.

### 2.2. Whole Genome Sequencing and Analysis

A PCR-free library was prepared from one Labrador Retriever case and sequenced in one lane of an Illumina HiSeq 4000 sequencer by GeneWiz (South Plainfield, NJ 07080, USA). Approximately 208 million 2 × 150 bp paired-end reads were generated, corresponding to 20-fold genome-wide coverage. The reads were mapped against the dog reference genome assembly (CanFam4) and the variants were called and annotated using the Ensembl Variant Effect Predictor (VEP) within our previously described bioinformatics pipeline [10,11]. Raw sequence reads are available in NCBI’s Short Read Archive at Permanent link: https://dataview.ncbi.nlm.nih.gov/object/SRR18391626 (accessed on 3 August 2022).

WGS variants obtained from the affected dog were compared to those of the control genomes from the University of Minnesota’s private WGS database containing 651 dogs of 63 diverse breeds (including 21 additional Labrador Retrievers from unrelated projects). Variants unique to the affected dog that were within or in close proximity to coding exons were prioritized as high (frame shift, loss or gain of stop or start codon, affecting a splice junction), moderate (missense) or low (synonymous, near splice junction) for further evaluation. A list of the coding variants is provided in Appendix A.

### 2.3. Haplotype Analysis

Haplotypes encompassing the *EHBP1L1* gene were evaluated within a cohort of 389 Labrador Retrievers previously genotyped via WGS, Axiom Canine Sets A and B or HD arrays (Thermo Fisher Scientific, Waltham, MA, USA) or the Illumina CanineHD BeadChip (Neogen, Lincoln, NE, USA). A total of 144 SNPs spanning 1 Mb that were common across platforms were extracted and phased with fastPHASE [12].

### 2.4. Variant Genotyping

The PCR primers flanking the reported *EHBP1L1* mutation reported in the present study were 5′-GCT TGT ACC ACC TTC ACG GA-3′ and 5′-CAC TGA CTA TCC TTC CCC GC-3′. The 171 bp product was submitted for Sanger sequencing and analyzed for the presence of the mutation utilizing Sequencer 5.1 software (Gene Codes Corporation, Ann Arbor, MI, USA). Genotyping of one affected Labrador for the *PTPLA* and *MTM1* variants associated with XLMTM and CNM in Labrador Retrievers was provided by a commercial laboratory.

## 3. Results

### Whole Genome Sequencing and Mutation Identification

WGS variants identified in the affected dog were initially compared to those of control genomes obtained from the University of Minnesota’s private WGS database [10,11]. This comparison generated a list of 16,684 variants unique to the case dog, including 499 variants residing within the coding regions of the genome. These coding sequence variants were prioritized as high (28), moderate (131) or low (340) priority based on their predicted effect on the gene’s or protein’s function, as described in the Materials and Methods Section, as well as denoted as being heterozygous (468) or homozygous (31) in the sequenced case. The complete list of all the initial 499 coding variants is presented in Appendix A.

We manually and visually interrogated the 4 high-priority and 15 moderate-priority variants present in homozygous form in the case. Three of these four high-priority variants and all 15 moderate-priority variants were ruled out based on low quality scores, an issue with library creation, annotation of the reference genome, or in one case, to be present in another WGS variant database [13,14]. In several cases, there were multiple variants clustering within the same read. The remaining nonsense variant (CFA18:52,128,140 G>A, XM_038563927.1 c.388C>T, p.R130*) residing within exon 5 of the *EHBP1L1* gene, was predicted to truncate greater than 90% of the protein, including conserved calponin-homology and bMERB domains (Figure 2). This variant was absent from more than 1000 dogs (including 38 Labrador Retrievers) across published WGS [13,14]. The literature relevant to selecting *EHBP1L1* as a biological candidate gene is reviewed in Section 4.

The affected full-sib of our WGS case dog, as well as two previously described ESS with a similar phenotype [1], were genotyped for the c.388C>T, p.R130* mutation. The full-sib Labrador Retriever was found to be homozygous for the *EHBP1L1* p.R130* variant, while the ESSs were both homozygous wild types. The pathological changes in the muscle biopsies of the *EHBP1L1* variant that the affected Labrador Retrievers were similar to those described for Labrador Retrievers with a centronuclear myopathy caused by mutations in *PTPLA* [9] and *MTM1* [8]. However, the affected Labrador was found to be clear for both known PTPLA and MTM1 variants.

Genotype data obtained from 144 SNPs surrounding the *EHBP1L1* p.R130* variant, spanning CFA18: 51,608,976–52,633,257, were extracted from a cohort of Labrador Retrievers and the phased haplotypes. Among these 389 Labrador Retrievers, 12 were observed to be heterozgyous for a haplotype indistinguishable from the case dog across the entire 1 Mb span (Figure 3). These 12 dogs, in addition to 2 dogs with similar haplotypes spanning less than the full 1 Mb region, were genotyped for the *EHBP1L1* variant. All 14 dogs genotyped as homozgyous wild types.

## 4. Discussion

Here, we describe a highly damaging *EHBP1L1* variant associated with dyserythropoietic anemia and polymyopathy in Labrador Retriever littermates. The information regarding the role of EHBP1L1 in human and animal disease is sparse. Nakajo et al. [15] reported that in epithelial cells deficient in Rab8, EHBP1L1, Bin 1 or dynamin, protein cargos are not transported to apical plasma membranes, but eventually accumulate in lysosomes. In the small intestines of Rab8a/b KO mice, the depletion of EHBp1L1 and Bin1 or inhibition of dynamin GTPase activity caused the missorting of apical proteins and abnormal intestinal villi [15]. Of interest, pathological changes in muscles of the Labrador Retrievers with mutant *EHBP1L1* included abnormal central localization of muscle nuclei and abnormal deposits of oxidative proteins revealed with mitochondrial-specific reactions (Figure 1). Such changes are also typically observed in centronuclear myopathy caused by variants in Bin1 [16] and dynamin 2 [17].

Furthermore, *EHBP1L1* KO mice were born alive, but died within one day after birth [15]. A post-mortem examination of the tissues showed no specific abnormalities, except for severe anemia determined by a histological examination. RBC morphology was not characterized further and muscles were not examined. A recent “molecular autopsy” study that included lethal variants, such as lethal malformation, still births and intrauterine fetal deaths, were identified in two families with *EHBP1L1* mutations with recurrent fetal loss and non-immune hydrops fetalis [18]. The pathological changes in the organs were not described, with the exception of abnormal intestinal villi as observed in the *EHBP1L1* KO mice [15]. From the above discussion, it appears that disease resulting from *EHBP1L1* mutations can be severe and have a broad spectrum of clinical signs that includes severe anemia and the likelihood of myopathy.

The *EHBP1L1* variant reported in the present study was the only variant from our WGS and filtering pipeline that merited serious consideration. It was unique to the case, present in homozygosity, truncated more than 90% of the protein and was also present in homozygosity in the other case, in sum yielding quite strong support for an association with the clinical syndrome. Additionally, when accompanied by the biological relevance discussed above, this suggests it being the causative variant. Although we were unable to perform an analysis to detect the presence of the EHBP1P1 protein or its fragments, we argue that a protein translated from any mRNA produced by the variant would be severely altered or totally nonfunctional.

That both affected sibs were homozygous is supportive, but of course not proof of autosomal recessive inheritance. Unfortunately, no additional family members were available to phenotype or genotype. Haplotypes encompassing the *EHBP1L1* gene that did not contain the mutation provided evidence that the mutation arose from a not uncommon haplotype within the Labrador Retriever breed. However, the mutation was not observed in a heterozygous form in our sampling of the related haplotypes, suggesting a very low allele frequency in the population.

A similar phenotype and pathological changes In red blood cells and myofibers were previously described in ESS dogs [1], but in this breed was also accompanied by concurrent cardiomyopathy. We genotyped two ESS dogs described in the original 1991 publication for the *EHBP1L1* c.388C>T, p.R130* variant and found them to be homozygous wild types. Further investigations of the basis of the condition in ESS dogs and the high likelihood of *EHBP1L1* involvement has been independently conducted by another research group [19]. Our report of an association of a mutation in the *EHBP1L1* gene with congenital dyserythropoietic anemia and polymyopathy provides new insights into the function of this gene and consequences resulting from its loss of function.

## 5. Conclusions

We described a highly damaging *EHBP1L1* missense variant associated with congenital dyserythropoietic anemia and polymyopathy in Labrador Retriever littermates. Our molecular genetic data, accompanied by descriptions of the gene’s biological relevance, yield strong support for it being the causative variant and may lead to new insights into the bases of dyserythropoiesis and polymyopathy.

## Figures and Tables

**Figure 1 genes-13-01427-f001:**
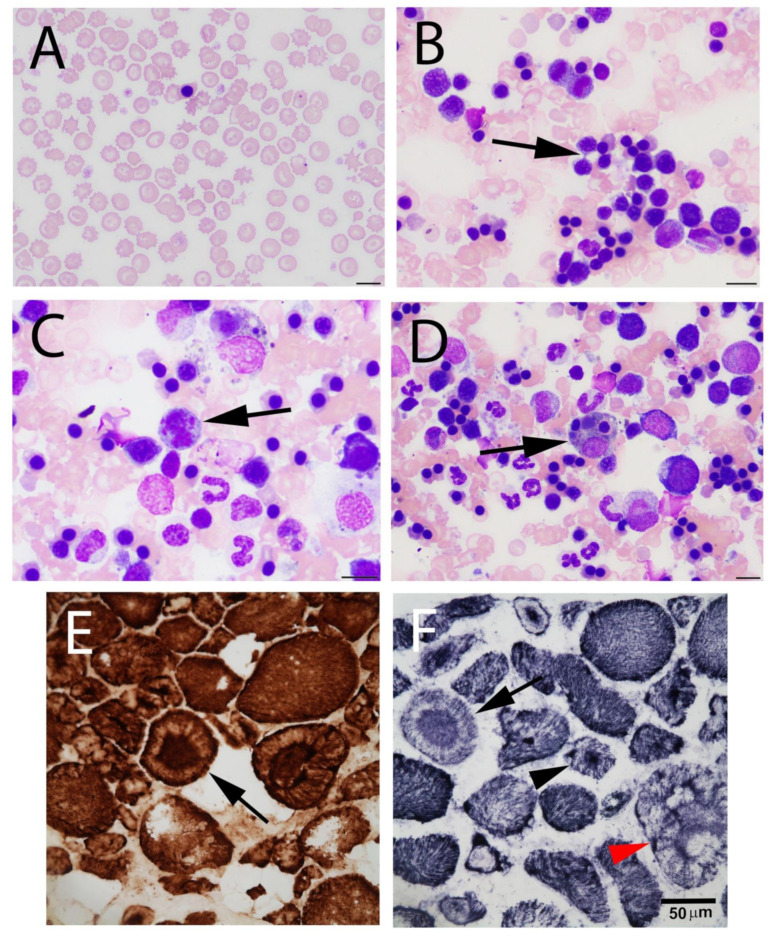
Review of histopathological changes in red blood cells, bone marrow and muscle in Labrador Retrievers with an *EHBP1L1* variant. (**A**) Representative photomicrograph of peripheral blood. Note the marked poikilocytosis characterized by schistocytes, microcytes and acanthocytes. A single metarubricyte is shown. (**B**) Representative photomicrograph of bone marrow findings displaying cytoplasmic bridging (arrow). (**C**) Representative photomicrograph of bone marrow displaying atypical mitotic figure (arrow). (**D**) Representative photomicrograph of bone marrow aspirate displaying rubriphagocytosis (arrow) and increased proportion of metarubricytes. Wrights Giemsa Stain used for (**A**–**D**). Bar = 10 µm for (**A**–**D**). Cryosections from the triceps were stained with the mitochondrial specific reactions cytochrome C oxidase (**E**) and succinic dehydrogenase (**F**). These images show variability in myofiber size, central accumulations of dark brown or dark blue reactivity (arrowheads, arrows) and central nuclei.

**Figure 2 genes-13-01427-f002:**
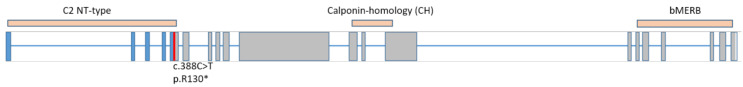
Structure of the *EHBP1L1* gene and site of the mutation. All 19 exons are filled in gray or blue. The premature stop codon mutation in exon 5 is marked by a red line. The wild-type protein will contain 19 exons (i.e., all blue and gray exons), while the mutant protein, if produced, would contain only exons 1–4 and part of exon 5 (i.e., the blue ones). The locations of the predicted homology domains, in reference to the human EHBP1L1 protein, are also indicated. A comparative amino acid sequence analysis determined a 70.3% identity between the canine and human orthologues.

**Figure 3 genes-13-01427-f003:**
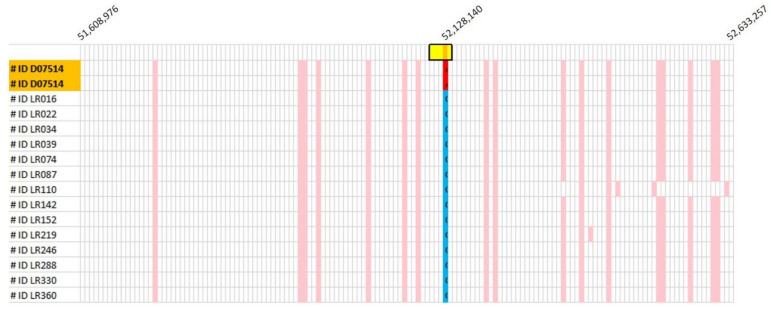
Examples of haplotypes encompassing the *EHBP1L1* gene. Haplotypes were generated from 144 SNPs within 1 Mb flanking the *EHBP1L1* gene. The gene location is denoted by a bolded yellow box at the top. Each row represents a haplotype obtained from the dog identified on the left. Each cell represents a SNP and the CFA18 positions of the 5′ and 3′ SNPs are indicated at the top; the mutation position is also indicated. The haplotype (highlighted in orange) that contains the *EHBP1L1* mutation (highlighted in red) is present as two copies (i.e., two rows) in the case. An identical 144 SNP haplotype was found as a single copy in twelve of the controls. These haplotypes were subsequently shown to differ from that in the case by having a reference allele (denoted as blue). Several other haplotypes containing a segment of the affected haplotype were also observed in the controls. These haplotypes also contained the reference allele.

## Data Availability

Raw sequence reads are available in NCBI’s Short Read Archive at Permanent link: https://dataview.ncbi.nlm.nih.gov/object/SRR18391626 (last accessed on 3 August 2022).

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
