# Peer review of "An EHPB1L1 Nonsense Mutation Associated with Congenital Dyserythropoietic Anemia and Polymyopathy in Labrador Retriever Littermates"

_genes, 2022, doi:10.3390/genes13081427_

Round 1

Reviewer 1 Report

The authors present a logical investigation associating a rare nonsense mutation in EHPB1L1 with a rare disorder in Labrador retrievers.  The follow on genetic studies are limited by lack of other samples from the affected family, but comparisons to other databases indicate that mutation is rare.  Overall the study is logical and clearly presented. My only suggestion is to add some additional information on the exclusion of the other homozygous high-impact variants.  For example, some additional information on the reason for concluding other variants were mapping errors, and what the cause of the error is (duplication? retrogene? Etc) would aid the reader and lend further support to the focus on the EHPB1L1 variant.

Author Response

We thank the reviewer for devoting time to this review and for kind comments.  In reply to the only suggestion we have added text (in red font) to the relevant paragraph in the Results section.

We manually and visually interrogated the 4 high priority and 15 moderate priority variants present in homozygous form in the case.  Three of these four high priority variants and all 15 moderate priority variants were ruled out based on low quality scores, an issue with library creation, annotation of the reference genome, or in one case, to be present in another WGS variant database [13,14]. In several cases there were multiple variants clustering within the same read. The remaining nonsense variant (CFA18:52,128,140 G>A, XM_038563927.1 c.388C>T, p.R130*), residing within exon 5 of the EHBP1L1 gene, is predicted to truncate greater than 90% of the protein, including conserved calponin-homology and bMERB domains (Figure 2). This variant was absent from more than 1,000 dogs (including 38 Labrador retrievers) across published WGS [13,14].  Literature relevant to selecting EHBP1L1 as a biological candidate gene is reviewed in the Discussion. 

Reviewer 2 Report

This is a succinct and clear manuscript that highlights the relevance of domestic dogs as a natural, comparative model for human disease. The authors detail a genomic case study in two litters of dogs affected by an inherited blood and muscle disorder. They characterize the clinical features of congenital dyserythropoietic anemia and polymyopathy in an affected litter of Labrador retrievers, and perform whole genome sequencing on samples from that litter, in addition to samples previously collected from a litter of affected English springer spaniels (ESS) in 1991. The work to generate whole genome sequence data for the affected pups is solid and leverages one of the latest high-contiguity canine reference assemblies. They compare genetic variants in affected dogs to those present in whole genomes from their institution’s private repository of 651 dogs, including 21 Labrador retrievers, to prioritize variants. While it might also be useful to broaden the set of control genomes to public repositories, especially for comparison of the affected ESS dogs to other ESS genomes, the authors may have chosen these comparison samples for known unaffected status, and a more thorough study of this disease and EHBP1L1 mutation in those ESS pups might be contained in the cited manuscript submitted to Genes by Østergård et al. They sufficiently demonstrate that the identified coding mutation of EHBP1L1 in the Labrador retriever litter is novel, discuss its relevance, and prioritize it as likely deleterious. The authors show that the mutation itself is unlikely to be widespread across Labrador retrievers, an analysis which has important relevance to the usage of discovered disease variants in canine genetic testing. The c.388C>T, p.R130* mutation has only been observed in this family of dogs and affected status not be detectable on lower density genotyping arrays given that the mutation is contained in a large haplotype indistinguishable from those in 389 Labrador retriever dogs with the wild type variant. These findings provide new support for the importance of EHBP1L1 dysfunction in the development of dyserythropoietic anemia and polymyopathy. 

Author Response

We thank the reviewer for devoting time to this review and for kind comments.